# Characterization of the Humoral Immune Response to Porcine Epidemic Diarrhea Virus Infection under Experimental and Field Conditions Using an AlphaLISA Platform

**DOI:** 10.3390/pathogens9030233

**Published:** 2020-03-21

**Authors:** Kay Kimpston-Burkgren, Juan Carlos Mora-Díaz, Philippe Roby, Jordan Bjustrom-Kraft, Rodger Main, Roger Bosse, Luis Gabriel Giménez-Lirola

**Affiliations:** 1Veterinary Diagnostic and Production Animal Medicine, College of Veterinary Medicine, Iowa State University, Ames, IA 50010, USA; kkb@iastate.edu (K.K.-B.); juanmora@iastate.edu (J.C.M.-D.); jnb@iastate.edu (J.B.-K.); rmain@iastate.edu (R.M.); 2Perkin Elmer, Waltham, MA 02451, USA; Philippe.Roby@perkinelmer.com (P.R.); Roger.Bosse@perkinelmer.com (R.B.)

**Keywords:** porcine epidemic diarrhea virus, humoral immune response, serum IgG, serum IgA, AlphaLISA

## Abstract

Coronavirus infections are a continuous threat raised time and again. With the recent emergence of novel virulent strains, these viruses can have a large impact on human and animal health. Porcine epidemic diarrhea (PED) is considered to be a reemerging pig disease caused by the enteropathogenic *alphacoronavirus* PED virus (PEDV). In the absence of effective vaccines, infection prevention and control through diagnostic testing and quarantine are critical. Early detection and differential diagnosis of PEDV infections increase the chance of successful control of the disease. Therefore, there is a continuous need for development of reduced assay-step protocols, no-wash, high-throughput immunoassays. This study described the characterization of the humoral immune response against PEDV under experimental and field conditions using a rapid, sensitive, luminescent proximity homogenous assay (AlphaLISA). PEDV IgG and IgA antibodies were developed toward the beginning of the second week of infection. PEDV IgG antibodies were detected for at least 16 weeks post-exposure. Remarkably, the serum IgA levels remained high and relatively stable throughout the study, lasting longer than the serum IgG response. Overall, AlphaLISA allows the detection and characterization of pathogen-specific antibodies with new speed, sensitivity, and simplicity of use. Particularly, the bridge assay constitutes a rapid diagnostic that substantially improves upon the “time to result” metric of currently available immunoassays.

## 1. Introduction

Porcine epidemic diarrhea virus (PEDV) is an *alphacoronavirus*, in the Coronaviridae family, of the order Nidovirales [1] that replicates in the cytoplasm of villous enterocytes of the small intestine and causes acute and watery malabsorptive diarrhea, vomiting, anorexia and dehydration in pigs [2,3]. PEDV has an enveloped virion, containing the single-stranded, positive-sense RNA genome of ~28 kb and seven open reading frames (ORFs) [4]. ORF1a and ORF1b encode the nonstructural replicase proteins required for transcription and translation. The structural proteins are encoded by the ORFs at the 3’ end of the genome, including the spike protein (S), the ORF3 accessory protein, the envelope protein (E), the membrane protein (M), and the nucleocapsid protein (N). The ORFs are flanked by 5’ and 3’ untranslated regions.

Original PEDV strains, referred to as classical PEDV strains, were first identified in Europe in the early 1970s [5], and spread to other European countries [6,7,8] and Asia in the late 1970s, causing sporadic and mild outbreaks in pigs of all ages through the 2000s [9,10,11,12]. PEDV evolved, and variants of highly virulent PEDV strains emerged in China in 2010 [13]. After its introduction to the United States in 2013 [14], and further spread to other countries in North, Central and South America [15], high morbidity and mortality has been reported mainly in neonates born to naïve sows. However, older pigs (feeders, finishers and adults) are also affected but with less overall severity and a higher rate of recovery than neonatal pigs [3,16]. Unfortunately, the age-related mechanisms associated to the severity of the disease have not been fully elucidated. To date, only one PEDV serotype has been recognized [17,18]. However, PEDV variants have different insertions or deletions (INDEL) in the S protein [19], which seems to be associated to differences on pathogenesis in conventional neonatal pigs [20].

PEDV enteric disease is clinically and histopathologically indistinguishable from other swine enteric pathogens including other porcine coronaviruses like transmissible gastroenteritis virus (TGEV) and porcine *deltacoronavirus* (PDCoV). Therefore, accurate PEDV diagnosis is dependent on laboratory methods. As reviewed by Diel et al. [21], standard tests for PEDV diagnosis include direct detection methods ranging from immunohistochemistry (IHC) to polymerase chain reaction (PCR) methods and indirect (antibody) detection methods such as enzyme-linked immunosorbent assay (ELISA), fluorescent focus neutralization (FFN), immunofluorescence assay (IFA), and fluorescent microbead-based immunoassay (FMIA).

Specifically for antibody detection, ELISA still remains the primary methodology of choice for diagnosis of infectious diseases across veterinary diagnostic laboratories. However, the immunoassay landscape has dramatically evolved in recent years looking for more analytically sensitive, rapid and simple, yet high-throughput screening methods. As an example of recent innovation, amplified luminescent proximity homogeneous assay (AlphaLISA) is a bead-based amplified luminescent proximity homogenous assay platform originally developed by PerkinElmer to study biomolecular interactions (Appendix A) [22]. The versatility and flexibility of this platform lies in that donor and acceptor beads can be coupled to antibodies and/or specific proteins which interact with the target analyte, bringing the beads into proximity with each other and leading to an energy transfer and emission of a chemiluminescent signal. Moreover, as a microfluidic non-wash platform, it allows a reduction in assay steps.

This study described the characterization of the antibody response against PEDV under experimental and field conditions using a rapid, sensitive AlphaLISA platform.

## 2. Materials and Methods 

### 2.1. Experimental Animal Study

The experimental animal study was carried out in the Iowa State University (ISU, Ames, IA, USA) Large Animal Research Facility (LAR). The study was approved by the Iowa State University Institutional Animal Care and Use Committee (5-15-8017-S) as part of a broader animal study described in detail in previous publications [23]. Briefly, twenty-four 7-week-old conventional pigs, pre-tested negative for PEDV, PDCoV, TGEV, porcine respiratory virus (PRCV) and porcine hemagglutinating encephalomyelitis virus (PHEV), were included in this study. Animals were divided into two groups (rooms) of 12 pigs each. One group was inoculated orally with 20 mL of PEDV non-S-INDEL strain (USA/IN/2013/19338E) at 1.0 × 10^6^ TCID_50_. The other group (control) was inoculated with 20 mL of Eagle’s minimum essential medium (EMEM, ATCC, Manassas, VA, USA). Serum samples (n = 264) were obtained from individual pigs at −7, 0, 3, 7, 10, 14, 17, 21, 28, 35, and 42 (euthanasia) days post-inoculation (dpi).

### 2.2. Field Animal Study

The field animal study was carried out in a commercial wean-to-finish barn in Missouri (USA), which consisted of 52 pens stocked with approximately 800 pigs placed at 3 weeks of age as previously described [24]. Six pens were selected (fixed) for serial sample collection. Five pig serum samples were collected from the same 6 pens starting at the day of placement, weekly for 6 weeks and biweekly thereafter for 19 weeks giving a total of 360 samples. Pigs were exposed to PEDV at 10 days post-placement (13-week-old) using water-mixed PEDV-positive material sprayed oronasally on pigs and feed.

### 2.3. Generation of S1 Recombinant Protein

A recombinant S1 glycoprotein was generated as described by Gimenez-Lirola et al. [23]. A consensus sequence of the amino-terminal S1 domain of the spike protein was generated from PEDV non-S-INDEL strains. The sequence was codon optimized for expression in mammalian cells. A 5’ terminal eukaryotic native signal was followed by a 3’ terminal tobacco etch virus cysteine protease site with a human IgG1 Fc portion tag. Genes were amplified then cloned in the pNPM5 expression vector (Novoprotein, Short Hills, NJ, USA), which was used to transfect into human embryonic kidney 293 (HEK293) cells (Invitrogen, Thermo Fisher Scientific, Grand Island, NY, USA) by polyethylenimine (PEI) treatment at a 1:4 plasmid to PEI ratio. Transfected cells were grown in serum free FreeStyle 293 expression medium (Gibco, Life Technologies, Carlsbad, CA, USA) at 37 °C with 5% CO_2_ and orbital shaking at 120 rpm. Five days after transfection, culture supernatants were harvested and filtered for subsequent purification. The expression of Fc tagged S1 protein was confirmed by sodium dodecyl sulfate polyacrylamide gel (12%) electrophoresis (SDS-PAGE). The Fc portion was then cleaved off and the S1 portion was purified by protein A chromatography and nickel-chelating Sepharose Fast Flow affinity chromatography (GE Healthcare, Pittsburgh, PA, USA) according to manufacturer instructions. Purified S1 protein (717 amino acids) was then dialyzed against phosphate buffered saline (PBS) pH 7.4, and analyzed by SDS-PAGE and Western blotting.

### 2.4. AlphaLISA Beads Conjugation

The S1 protein was conjugated to raw (aldehyde) AlphaLISA Acceptor and Donor beads via reductive amination and according to manufacturer’s protocol (PerkinElmer Health Sciences, Inc., Boston, MA, USA). Briefly, 0.1 mg of S1 protein (1 mg/mL) was mixed with 1 mg AlphaLISA beads in a 2 mL vial (Eppendorf, Hauppauge, NY, USA) and completed with reaction buffer (100 mM HEPES pH 7.4) to a final reaction volume of 200 µL. Then, 1.25 µL of 10% Tween-20 (Sigma, St. Louis, MO, USA) and 10 µL of a 400 mM solution of the reducing agent sodium cyanoborohydride (NaBH3CN; Sigma) were added to the 2 mL vial and the reaction mix was incubated for 18–24 h at 37 °C with mild agitation (6–10 rpm) using a rotary shaker. Unreacted aldehyde groups on beads were blocked by adding 10 µL solution of carboxymethoxylamine (CMO, Sigma) at 65 mg/mL in 800 mM sodium hydroxide (NaOH, Sigma) followed by a 60 min incubation at 37 °C under mild agitation. The uncoupled S1 protein was removed by centrifugation (16,000× *g* for 15 min at 4 °C) and washing the beads three times. Supernatant was removed with a micropipette and the bead pellet resuspended in 200 µL of 100 mM Tris-HCl pH 8.0. The bead solution was briefly sonicated in a sonicating water bath (Branson Ultrasonic Corp., Danbury, CT, USA) to ensure that the beads are not aggregated. After the last centrifugation, beads were resuspended at 5 mg/mL in storage buffer (200 µL of PBS + 0.05% Proclin-300 as a preservative), re-sonicated and stored at 4 °C until use.

### 2.5. AlphaLISA IgG/IgA Antibody Assay

This AlphaLISA procedure was designed as a 2 h, two-step, no-wash, isotype-specific (IgG, IgA) immunoassay, where PEDV IgG and/or IgA antibody could be detected simultaneously, in two separate wells, within the same plate (Figure 1A). Incubations were performed in half area 96-well white plates (PerkinElmer) at room temperature (~22 °C), protected from light, and on a rotating shaker. In the first step either 5 µL of serum sample or control, pre-diluted 1:100 in PBS pH 7.4 (Gibco, Thermo Scientific), was incubated for 1 h with 20 µL of a mix containing 25 µg/mL of S1 protein-conjugated AlphaLISA Acceptor beads and 7.5 nM biotinylated goat anti-pig IgG (Fc) or 5 nM biotinylated goat anti-pig IgA (Bethyl Laboratories Inc., Montgomery, TX, USA) in assay buffer 1x (AlphaLISA Assay Buffer 10X; PerkinElmer). Then, in a second step, 25 µL of 80 µg/mL Streptavidin Donor beads (PerkinElmer) were directly added to each well and incubated for 1 h in a final reaction volume of 50 µL per well. In the presence of PEDV-specific IgG and/or IgA antibodies, Acceptor and Donor beads come into close proximity. Laser irradiation of Donor beads at 680 nm caused the release of singlet oxygen molecules that triggered a cascade of chemical events in nearby Acceptor beads, which resulted in a sharp peak in chemiluminescent emission at 615 nm that was read in an EnVision^®^ multimode plate reader (PerkinElmer) (Appendix A). AlphaLISA results were converted into sample-to-positive ratios on the basis of the positive and negative internal controls.

### 2.6. AlphaLISA Total Antibody Bridging Assay

This AlphaLISA bridge assay was designed as an antigen sandwich immunoassay for rapid detection (5 to 60 min, 1-step) of the total antibodies specific for PEDV (Figure 1B). In brief, 5 µL of undiluted serum samples or controls were incubated with 45 µL of bead mix containing 20 µg/mL of both S1-coupled Donor and S1-coupled Acceptor beads in assay buffer for 5, 30 or 60 min at room temperature (~22 °C), preserved from light, and on a rotating shaker. Results were obtained and processed as previously described for the isotype-specific assay.

### 2.7. Data Analysis

To assess the differences in the PEDV antibody responses between experimentally inoculated PEDV vs. control groups, a generalized linear model with mixed effects (PROC GLIMMIX) was performed for IgG and IgA separately. Treatments (PEDV vs. medium inoculation) and dpi were fixed effects, while the pigs were the random effect due to the repeated measurements on the same animals. A *p*-value < 0.05 was considered statistically significant. Pearson correlation analysis between the IgG and IgA response (daily average of AlphaLISA S/P values) in PEDV inoculated pigs was performed by dividing the experimental study (49 days) into three periods: −7 to 3 dpi, 7 to 17 dpi, and 21 to 42 dpi. To determine the difference in S/P values among the three incubation periods for the PEDV inoculated group, a GLIMMIX model was used with incubation time and dpi as the fixed effects and the pigs as the random effect. 

The PEDV isotype-specific (IgG and IgA) antibody response under field conditions was evaluated using a GLIMMIX model, where antibody isotype and week post-exposure (wpe) were the fixed effects, and the pigs were the random effect. Pearson correlation analysis between the IgG and IgA responses (weekly average of AlphaLISA S/P values) in PEDV inoculated pigs was performed by dividing the field animal study (19 weeks) into three periods: −2 to 1 wpe, 2 to 8 wpe, and 10 to 16 wpe.

Optimal cutoff values and diagnostic performance of the PEDV AlphaLISA platform were estimated by receiver operating characteristic curve (ROC) analysis. All statistical analyses were performed using SAS version 9.4 (SAS, Cary, North Carolina/USA, SAS Institute, Inc.).

## 3. Results

### 3.1. Diagnostic Performance of AlphaLISA IgG/IgA Antibody Assay

A ROC analysis was performed for both AlphaLISA IgG and IgA assay. Selected S/P cutoff values were 0.25 for IgG and 0.15 for IgA. The PEDV antibody-positive detection rates per day post-inoculation or exposure under experimental or field conditions are presented in Table 1 or Table 2, respectively. The overall diagnostic specificity, calculated by using over 257 negative serum samples, was 99.2% for IgG and 100% for IgA.

### 3.2. PEDV Isotype-Specific Antibody Response under Experimental Conditions

Pigs in the negative control group remained IgG/IgA seronegative throughout the study. PEDV-inoculated pigs developed a detectable IgG antibody response by 10 dpi. The IgG response increased overtime, reaching a plateau by dpi 35. PEDV IgG response was statistically significant (*p* < 0.05), compared to the negative control group, from dpi 14 until the end of the study (Figure 2A).

Although PEDV IgA response was first detected by 7 dpi (Table 1), the average IgA response was statistically significant (*p* < 0.05) compared to the negative group from dpi 10 to 42. The IgA S/P ratios increased until 14 dpi, decreased slightly at 17 and 21 dpi, and continued to rise until the end of the study. (Figure 2B).

Overall, the IgG and IgA S/P values were positively correlated (0.89) throughout the experimental study. Specifically, the correlation for the first period (dpi −7 to 3) was 0.96, 0.64 for the second period (dpi 7 to 17), and 0.97 for the third period (dpi 21 to 42). 

### 3.3. PEDV Isotype-Specific Antibody Response under Field Conditions

The isotype specific antibody response to PEDV exposure under field conditions is presented in Figure 3. PEDV IgG and IgA antibodies were detected between 1–2 weeks post-exposure. IgG response decreased consistently thereafter, while the IgA response peaked ~8 weeks post-exposure and remained fairly stable until the end of the study.

Overall, correlation analysis comparing IgG and IgA responses under field conditions showed a low positive correlation (0.67) over time. A weak positive correlation was observed for the first (−2 to 1 wpe; 0.62) and third (10 to 16 wpe; 0.40) periods. Contrary, a negative correlation between IgG and IgA antibody response (opposite trend) was observed for the second period (2 to 8 wpe; 0.92).

### 3.4. Rapid detection of PEDV Total Antibodies by AlphaLISA Bridging Assay

The PEDV total antibody bridging rapid assay was evaluated on the experimental samples. Antibody response at three different incubation periods were compared: 5, 30, and 60 min. PEDV antibody response was detected, and peaked, at 7 dpi, decreased until 17 dpi, being stable thereafter (Figure 4). No antibody response was detected in the negative control group. No significant differences (*p* > 0.05) on total antibody S/P values were found, regardless of incubation time.

## 4. Discussion

The emergence of highly virulent PEDV strains in the United States in 2013 fostered the development of a greater number of diagnostic tools broadly described through the literature. While PEDV RNA detection in fecal specimens is used to confirm presence of PEDV during outbreaks of severe diarrhea in neonatal pigs with high mortality, serology tests are important for surveillance, detection of asymptomatic or mild cases, monitoring response to immunization strategies, and demonstrating previous exposure to the virus. However, there is a continuous need for development of reduced assay-steps protocols and no-wash immunoassays allowing to run more samples in a shorter period of time. Thereby, the recent development of bead-based methods and microfluidic platforms like AlphaLISA have started to impact immunoassays, taking the sensitivity of traditional enzyme immunoassays to the next level.

A recombinant protein derived of the amino-terminal portion (S1) of the PEDV spike protein was used as an antigen for the development of the AlphaLISA immunoassay platforms described herein. In brief, the homotrimeric S glycoprotein, class I fusion protein essential for both host specificity and viral infectivity, is the main driver of the neutralizing antibody and T-cell immune responses against PEDV and a major immunogenic target for vaccine design [25,26,27]. Similar to other coronaviruses, PEDV S protein is composed of two subunits; the highly variable S1 subunit (N-terminal) contains receptor-binding domain(s), which remains unknown, responsible for viral attachment and the conserved S2 subunit (C-terminal) which mediates fusion between the viral and host cell membranes, and contains immunodominant neutralizing epitopes [28,29]. The selected S1 region is characterized by multiple amino acid insertions, deletions, and substitutions [30], that allow for a differential serodiagnosis of PEDV and rule out cross-reactivity against other porcine coronaviruses that might be cocirculating in swine herds [23]. However, S1 may not recognize some neutralizing antibodies specifically directed against S2 [29]. Although structural proteins N and M are highly antigenic, they are also one of the most conserved and, therefore, cross-reactive proteins among coronaviruses [23].

There is a need for research and development to translate new technologies into practical tests. Therefore, the aim of the study was to characterize the humoral response to oral exposure to PEDV non S-INDEL strain, currently circulating in North America, under controlled (i.e., time to exposure to PEDV) experimental (livestock-infectious disease BSL-2 isolation facility) and field (wean-to-finish commercial farm) conditions, respectively. To achieve this objective, a homogeneous (no washing steps required) 2 h and two-steps AlphaLISA IgG/IgA assay platform was used. Experimental serum samples were previously tested by IgG ELISA and multiplex Luminex platform [24], while field serum samples were previously tested by IgG/IgA ELISA [23]. Like in these previous studies, PEDV IgG and IgA specific antibodies were detected by AlphaLISA toward the beginning of the second week of infection, regardless of exposure conditions. Longitudinal field study allowed further assessment of the duration of IgG and IgA antibody response in a serum. In concordance with our previous study [24], PEDV IgG antibodies were detected for at least 16 wpe by AlphaLISA, showing a similar downward trend than that obtained by ELISA. However, the AlphaLISA serum IgA levels remained high and relatively stable throughout the study lasting longer than the IgA levels previously reported by ELISA. Unexpectedly and contrary to the serum antibody profile obtained by ELISA, we reported a strong long-lasting AlphaLISA IgA response compared to the AlphaLISA IgG response. This study demonstrates that isotype antibody detection would depend on the analytical sensitivity and dynamic range of the immunoassay platform used.

Overall AlphaLISA enables the detection and characterization of pathogen-specific antibodies with new speed, sensitivity, and simplicity of use. Particularly, the bridge assay constitutes a rapid diagnostic that substantially improves upon the “time to result” metric of currently available immunoassays.

## Figures and Tables

**Figure 1 pathogens-09-00233-f001:**
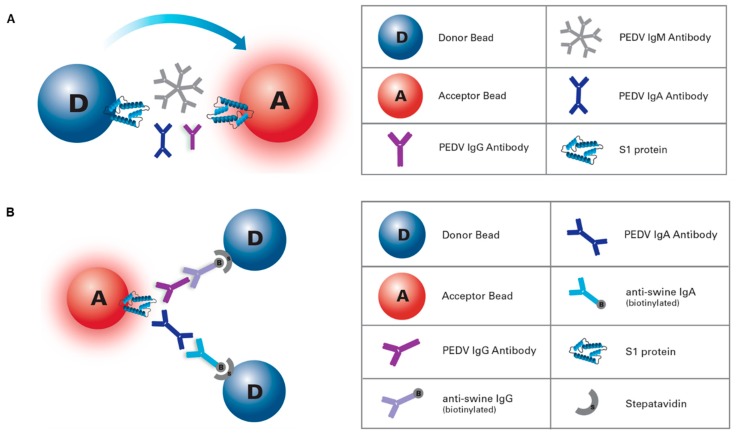
Porcine epidemic diarrhea virus (PEDV) amplified luminescent proximity homogeneous assay (AlphaLISA) platforms. (**a**) PEDV AlphaLISA IgG/IgA antibody assay: 2 h, two-steps, and no-wash, isotype-specific immunoassay, where PEDV IgG and/or IgA antibody can be detected simultaneously, in two separate wells, within the same plate. (**b**) AlphaLISA total antibody bridging assay: double antigen sandwich immunoassay for rapid detection of PEDV total antibodies.

**Figure 2 pathogens-09-00233-f002:**
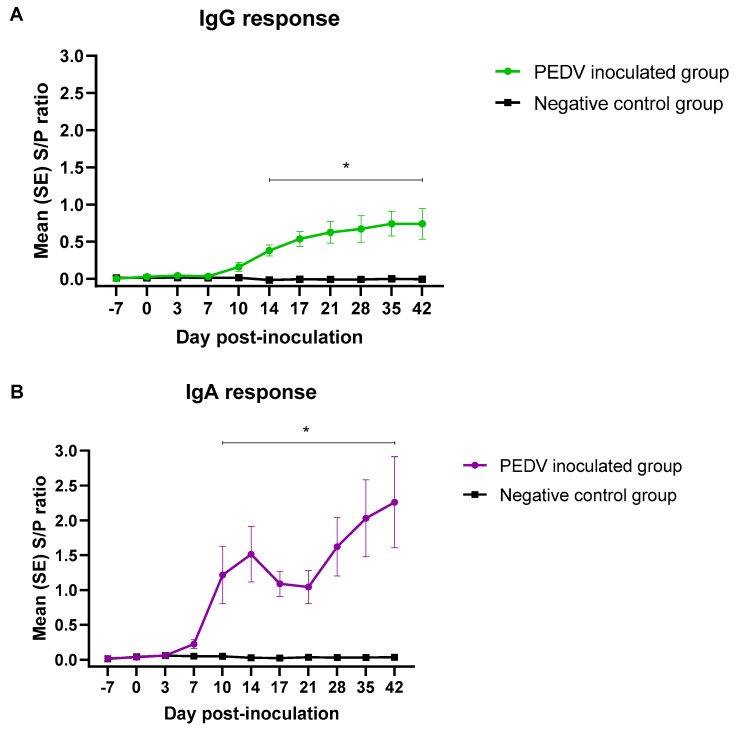
AlphaLISA PEDV isotype-specific IgG (**A**) and IgA (**B**) antibody responses (mean S/P values, SE) overtime (49 days) in pigs (n = 12) inoculated with PEDV or mock inoculated (n = 12) with culture medium under experimental conditions. *Denoted statistical differences (*p* < 0.05).

**Figure 3 pathogens-09-00233-f003:**
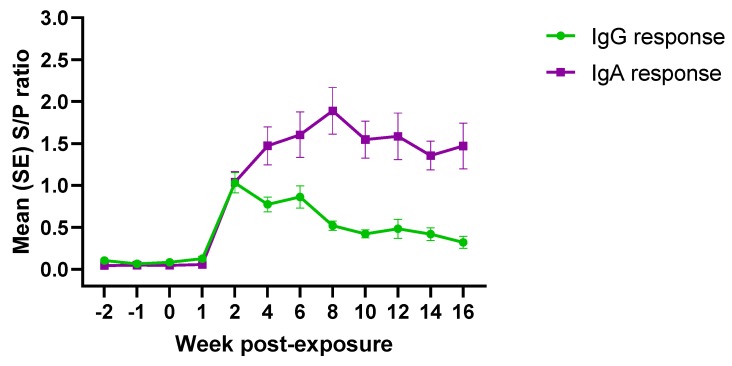
AlphaLISA PEDV isotype-specific (IgG/IgA) antibody responses (mean S/P values, SE) over time (19 weeks) in pigs (n = 30) exposed to PEDV under field conditions. *Denoted statistical differences (*p* < 0.05).

**Figure 4 pathogens-09-00233-f004:**
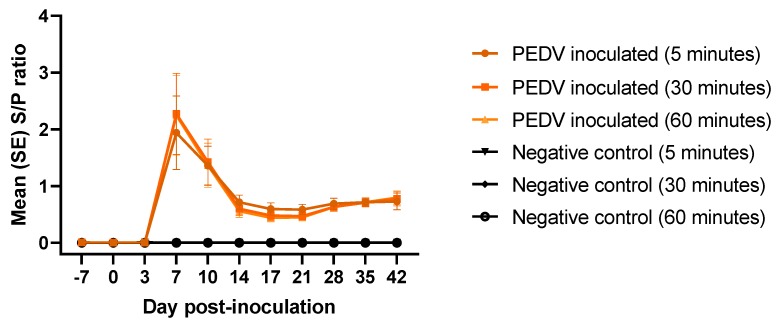
Rapid detection of PEDV total antibodies (average S/P, SE) at different incubation times (5, 30, and 60 min) on serum samples from pigs (n = 12) experimentally inoculated with PEDV or mock inoculated (n = 12) with culture medium using AlphaLISA bridging rapid assay.

**Table 1 pathogens-09-00233-t001:** PEDV-positive detection rate per day post-inoculation under experimental conditions.

Response	Group	Day Post-Inoculation										
		−7	0	3	7	10	14	17	21	28	35	42
IgG	Negative	0/12	0/12	0/12	0/12	0/12	0/12	0/12	0/12	0/12	0/12	0/12
	PEDV	0/12	0/12	0/12	0/12	1/12	9/12	9/12	9/12	10/12	11/12	11/12
IgA	Negative	0/12	0/12	0/12	0/12	0/12	0/12	0/12	0/12	0/12	0/12	0/12
	PEDV	0/12	0/12	0/12	7/12	10/12	11/12	10/12	10/12	11/12	11/12	11/12

**Table 2 pathogens-09-00233-t002:** PEDV-positive detection rate per week post-exposure under field conditions.

Response	Week Post-Exposure											
	−2	−1	0	1	2	4	6	8	10	12	14	16
IgG	0/30	1/29	1/30	4/30	28/30	27/30	24/27	26/30	20/30	19/30	20/30	13/30
IgA	0/30	0/29	0/30	2/30	29/30	28/30	25/27	30/30	28/30	29/30	30/30	28/30

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
