# Peer review of "Characterization of the Humoral Immune Response to Porcine Epidemic Diarrhea Virus Infection under Experimental and Field Conditions Using an AlphaLISA Platform"

_pathogens, 2020, doi:10.3390/pathogens9030233_

Round 1

Reviewer 1 Report

In this manuscript “Characterization of the humoral immune response to porcine epidemic diarrhea virus infection under experimental and field conditions using an AlphaLISA platform”, Kimpston-Burkgren et al. using AlphaLISA to assay pig IgG and IgA antibodies developed by PEDV infection under experimental and field conditions. The authors provided a well-organized set of experimental data showing that AlphaLISA is a rapid and sensitive method for the characterization of PEDV-specific antibodies.

Major comments:

  1. Although the results of PEDV-specific serum detection and quantification using AlphaLISA are compelling, authors should compare the results with conventional ELISAs to claim that AlphaLISA is a fast and accurate method for detecting PEDV antibodies.
  2. The authors claimed that the serum IgA levels remained high and relatively stable throughout the study and lasted longer than the serum IgG response. They should compare the levels of IgM because the technology to detect this antibody is available and it is one of the major antibodies raised after the infection with the viruses.

Minor comment:

The spike is a trimeric protein but not S1 domain used in this experiment. Therefore, although AlphaLISA using the S1 domain works well, it may not recognize some of the neutralizing antibodies. This can be discussed at the end of this article.

Author Response

March 12, 2020

Re: Manuscript ID# pathogens-743818

Dear Reviewer,

We thank you for your comments. We feel they have resulted in an improved manuscript and appreciate your efforts on our behalf. Line numbers listed below refer to the edited manuscript with "markup" option turned on.

Sincerely,

Dr.  Luis G. Giménez-Lirola

Comments and Suggestions for Authors

In this manuscript “Characterization of the humoral immune response to porcine epidemic diarrhea virus infection under experimental and field conditions using an AlphaLISA platform”, Kimpston-Burkgren et al. using AlphaLISA to assay pig IgG and IgA antibodies developed by PEDV infection under experimental and field conditions. The authors provided a well-organized set of experimental data showing that AlphaLISA is a rapid and sensitive method for the characterization of PEDV-specific antibodies.

Response: Thank you for your comments!.

Major comments:

Although the results of PEDV-specific serum detection and quantification using AlphaLISA are compelling, authors should compare the results with conventional ELISAs to claim that AlphaLISA is a fast and accurate method for detecting PEDV antibodies.

Response:  Both experimental and field serum samples used in this study were previously tested by ELISA and reported in previous publications. However, the manuscript has been reviewed to improve the discussion on this regard (Line 302-313).

The authors claimed that the serum IgA levels remained high and relatively stable throughout the study and lasted longer than the serum IgG response. They should compare the levels of IgM because the technology to detect this antibody is available and it is one of the major antibodies raised after the infection with the viruses.

Response: We agree with the reviewer on the overall relevance of IgM antibodies produced in response to different viruses during acute phase post-infection. However, specifically for PEDV, it has been demonstrated that the IgG and IgA antibody responses are more relevant against PEDV infection. Recent studies support our observations, e.g., Lin et al (BMC Vet Res 2019) reported no detectable serum PEDV-specific IgM antibody response in conventional weaned pigs.

Minor comment:

The spike is a trimeric protein but not S1 domain used in this experiment. Therefore, although AlphaLISA using the S1 domain works well, it may not recognize some of the neutralizing antibodies. This can be discussed at the end of this article.

Response: The discussion has been reviewed to address this concern (Line 289; Line 292-293).

Reviewer 2 Report

This manuscript describes about AlphaLISA, new immunoassay technique to detect PEDV rapidly.

However, the major finding in this study should is application example of the known technique.

This manuscript has a few novel findings, and hence this manuscript needs to be improved according to below comments.

Major comments:

The style of manuscript should be changed article to Brief report.

The author should compare and discuss about onsets of detection not assay steps between your method and other immunoassay (neutralization test and immunofluorescence assay) in detail.

The author has better add the results of this assay for animals infected with avirulent PEDV.

Author Response

March 12, 2020

Re: Manuscript ID# pathogens-743818

Dear Reviewer,

We thank you for your comments. We feel they have resulted in an improved manuscript and appreciate your efforts on our behalf. Line numbers listed below refer to the edited manuscript with "markup" option turned on.

Sincerely,

Dr.  Luis G. Giménez-Lirola

Comments and Suggestions for Authors

This manuscript describes about AlphaLISA, new immunoassay technique to detect PEDV rapidly.

However, the major finding in this study should is application example of the known technique.

This manuscript has a few novel findings, and hence this manuscript needs to be improved according to below comments.

Response: Thank you for your comments!.

Major comments:

The style of manuscript should be changed article to Brief report.

Response: The discussion has been significantly reduced following reviewer’s advice. We also suggest to move previous figure 1 to supplementary material (Supplementary figure S1). If accepted, we would like that the editor decide on the final format of the manuscript.

The author should compare and discuss about onsets of detection not assay steps between your method and other immunoassay (neutralization test and immunofluorescence assay) in detail.

Response: The discussion has been extensively reviewed to address this issue (Line 289; Line 292-293).

The author has better add the results of this assay for animals infected with avirulent PEDV.

Response: In this study, we used a PEDV non S-INDEL strain originally isolated and described during original PEDV outbreaks in the United States. These strains was selected because its representative of the virulent strains currently circulating in North American and some countries of Central and South America. ). Overall, S-INDEL strains cause lower neonatal mortality than the high-virulence non-S INDEL strains. However, none of the animals used in both experimental and field studies died during our experiments. No other PEDV strain was used in this particular study but additional studies towards the characterization of the immunopathogenesis of moderate vs. high virulent PEDV strains in different groups of age are currently ongoing.

As additional information, we reported that antibodies produced against the U.S. PEDV non-S-INDEL strain used in this study reacted similarly to S1 proteins derived from either S-INDEL and non-S-INDEL PEDV strains (Gimenez-Lirola et al. J Clin Microbiol 2017). We also demonstrated that the antibodies against U.S. PEDV S-INDEL and non-S-INDEL strains cross-reacted and cross-neutralized both strains in vitro (Chen et al. BMC Vet Res 2016).

Round 2

Reviewer 1 Report

 I agree with the author's answers and have no further comment on the publication of this manuscript. 

Author Response

Thank you for reviewing this manuscript!

Reviewer 2 Report

no comments

Author Response

(The authors gave the same response as above.)
